# Contribution of Attenuation of TNF-α and NF-κB in the Anti-Epileptic, Anti-Apoptotic and Neuroprotective Potential of *Rosa webbiana* Fruit and Its Chitosan Encapsulation

**DOI:** 10.3390/molecules26082347

**Published:** 2021-04-17

**Authors:** Anum Firdous, Sadia Sarwar, Fawad Ali Shah, Sobia Tabasum, Alam Zeb, Humaira Nadeem, Abir Alamro, Amani Ahmed Alghamdi, Arooj Mohsin Alvi, Komal Naeem, Muhammad Sohaib Khalid

**Affiliations:** 1Departament of Pharmacognosy, Riphah Institute of Pharmaceutical Sciences, Faculty of Pharmaceutical Sciences, Riphah International University, Islamabad 44000, Pakistan; anumfirdous23@gmail.com (A.F.); muhammadsohaibkhalid86@gmail.com (M.S.K.); 2Departament of Pharmacology, Riphah Institute of Pharmaceutical Sciences, Faculty of Pharmaceutical Sciences, Riphah International University, Islamabad 44000, Pakistan; fawad.shah@riphah.edu.pk (F.A.S.); aroojalvi@hotmail.com (A.M.A.); komal.naeem@riphah.edu.pk (K.N.); 3Departament of Biological Sciences, International Islamic University, Islamabad 44000, Pakistan; sobia.tabasum@iiu.edu.pk; 4Departament of Pharmaceutics, Riphah Institute of Pharmaceutical Sciences, Faculty of Pharmaceutical Sciences, Riphah International University, Islamabad 44000, Pakistan; alam.zeb@riphah.edu.pk; 5Departament of Pharmaceutical Chemistry, Riphah Institute of Pharmaceutical Sciences, Faculty of Pharmaceutical Sciences, Riphah International University, Islamabad 44000, Pakistan; humaira.nadeem@riphah.edu.pk; 6Departament of Biochemistry, College of Science, King Saud University, P.O. Box 22452, Riyadh 11495, Saudi Arabia; aalamro@ksu.edu.sa (A.A.); aalghamedi@ksu.edu.sa (A.A.A.)

**Keywords:** anticonvulsant, Rosaceae, *n*-hexane extract

## Abstract

*Rosa webbiana* L. (Rosaceae) is one of the least reported and most understudied members of this family. It is native to the Himalayan regions of Pakistan and Nepal. The anti-convulsant effect of *n*-hexane extract of fruit of *Rosa webbiana* was investigated in a pentylenetetrazole (PTZ)-induced animal model of epilepsy. Male Sprague-Dawley rats were divided into six groups (n = 7) including control, PTZ (40 mg/kg), diazepam (4 mg/kg) and *n*-hexane extract (at 50, 150 and 300 mg/kg). Convulsive behavior was observed and resultant seizures were scored, animals sacrificed and their brains preserved. Chitosan nanoparticles were prepared using the ionic gelation method and characterized by UV-analysis, zeta potential and Fourier transform infrared spectroscopy (FTIR). The effects of all the treatments on the expression of phosphorylated cytokine tumor necrosis factor α (p-TNF-α) and phosphorylated transcription factor nuclear factor kappa B (p-NF-κB) expression in the cortex and hippocampus of the brains of treated rats were studied through enzyme linked immunosorbent assay (ELISA) and morphological differences and surviving neuronal number were recorded through hematoxylene and eosin (H&E) staining. Significant changes in seizures score and survival rate of rats were observed. Downregulation of neuro-inflammation, p-TNF-α and p-NF-κB was evident. Gas Chromatography-Mass Spectrometry (GC-MS) analysis of this fraction showed multiple constituents of interest, including esters, alkanes and amines.

## 1. Introduction

Neurodegenerative diseases pose a substantial threat and cover a wide range of neuronal diseases. Continuous damage to the structure of neurons in the brain leads to the loss of their functional activity, ultimately culminating in the death of neuronal cells [1]. More effective treatment options are needed to be explored to combat the devastating neuronal disorders [2]. Epilepsy is a chronic condition affecting predominantly the young population, being 5% of all neuronal disorders reported worldwide annually [3]. Excitatory glutamatergic mechanisms, inflammation and oxidative stress are involved in seizures associated with epilepsy. Inflammation in brain hyperexcites the neurons and dysregulates the immune-inflammatory function, which contributes to seizures. Concurrently, seizures promote the production of pro-inflammatory cytokines and downstream events, which reinforce each other in the form of vicious positive feedback loop. NF-κB is evidently a main factor involved in transcription-dependent cellular changes in neuronal cells and modulates the behavior and cognition by activating different genes [4]. This is one of the inflammatory molecules whose levels determine and control the neuromodulation pathway and plays a role in the hyperexcitation of the neuronal matrix, which accentuates seizures. Inhibition of certain inflammatory markers can cease the epileptic seizures [5]. There are numbers of cytokines which are produced by the immune system in response to neuronal insults. Tumor necrosis factor (TNF-α) is a pro-inflammatory cytokine and controls the glutamate receptors trafficking. TNF receptor 1 (TNFR1) and TNF receptor 2 (TNFR2), thus, can excite the neurons by increasing glutamate level in synaptic cleft. A previous study employing kindling model of epilepsy has reported a correlation between TNF-α and epileptogenesis, where TNF-α administered rats exhibited prolonged seizure [6].

The use of plant-based products to treat the convulsions is an ancient practice. Plants have been traditionally used to treat epilepsy in China, Europe, Iran and Egypt. Plants and herbal medicine are in practice in these different cultures as a first line therapy and adjuncts. Among 25 herbal medicines mentioned in traditional medicine books of Iran, *Paeonia officinalis*, *Ferula asafetida*, *Coriandrum sativum*, *Cedrus deodara loudon*, *Lavandula stoechas*, *Ferula persica* and *Caesalpinia bonducella* Roxb are worth mentioning. *Ferula asafetida*, *Lavandula officinalis* and *Gummosa boiss* control the PTZ induced seizure in animal model through their antioxidant effect, enhancing the release of GABA and by inhibiting glutamate release. The Ayurveda, Sidha and Unani system of medicines are the most famous traditional systems in India, having existed for centuries. An important herb that belongs to the Indian Traditional system of medicines, studied against Epileptogenesis, is *Zizphus jujube* [7,8]. In another study, five different herbal medicines that were used in old ages to treat the epilepsy, namely, Makhzan ul-Adwia, Al-abniah, Tuhfat al Mu’ minin, Haqaeq al Adwia and canon, were reported. Similarly, “Himalayan yew” is used by natives of the northern region of Nepal and Pakistan to treat epileptic patients [9]. A number of active compounds isolated from Asian herbs have been investigated by the National Institute of Neurological Disorders and Stroke (NIH-NINDS) under the Harvard program of epilepsy. Huperzine A was isolated from Chinese club, which is a lycopodium alkaloid, and it acts by blocking the N-methyl-D-aspartic acid (NMDA) receptor, which mimics the glutamate action [8]. Several species of the family Rosacea, including *Rosa canina* and *Rosa damascene*, are well reputed for their role in mental health care [9]. *Rosa webbiana* is a close relative of these species and is often deliberately or undeliberately marketed as *Rose hip* (*Rosa canina*). We are investigating this species for multiple pharmacological activities and the different extracts of fruit has shown anticancer (in estrogen-mediated cell lines), antiestrogenic, hepatoprotective and chemoprotective effects, but results are yet not published as the work is in progress. We could not find sufficient literature evidence either on phytochemical or pharmacological profile of this species of Rosacease. It is distributed along Himalayan range, in the Turmic Valley Karakaram National Park, and in the Shigar Valley Gilgit-Baltistan (3000–3600 m). In these areas, it is a traditional remedy for hypertension, cold, flu, blood purification, mental relaxation and for skin problems [10,11,12].

The use of nanocarriers to address the issues of herbal preparations related to their low bioavailability, poor stability, low solubility, poor permeability, low absorption and biodistribution and rapid elimination has been appreciably increased in the last two decades. Multiple mechanisms have been explored which improve the oral absorption and bioavailability of herbal drugs including increased solubility and dissolution rate, enhanced protection against rapid degradation, enhanced lymphatic absorption, higher drug permeation, resistance to different metabolic processes and longer systemic circulation time [13]. For the delivery of neuromedicine, the largest hurdle is the Blood–Brain Barrier (BBB), which in normal circumstances protects the brain from foreign substances and systemic enzymes, allowing only selective molecules to pass through [14]. Nanoparticles (NPs) are ideal candidates to overcome all the hindrance of BBB because of their small size (10–1000 nm), concordant morphology and flexible surface characteristics and huge loading capacity [15]. There are countless reports in the literature where different nanomaterials, including carbon-based nanoparticles, metal-based nanoparticle, polymers of natural or synthetic origin and lipid-based and semiconductor nanoparticle, have been employed [16]. The advantage of using natural polymer encapsulation is their compatibility with human body and their complex matrix structure which can control the release and number of active agents [17]. Fruit of *Rosa webbiana* is edible and is popular among local children, indicating its safety. It has also been reported to be rich in vitamin C [18,19]. This evidence encouraged us to design our experiment to investigate the anti-convulsant potential of *Rosa webbiana* fruit so that this high-potential and rich nutritional source can be exploited as an alternative healthcare option. Currently, the fruit normally goes to waste as the shrub is used as fuel by local inhabitants, facing the danger of extinction. 

## 2. Results

### 2.1. Anti-Convulsant Effect and Cytotoxicity of N-Hexane, Ethyl Acetate and Methanol Extracts of Fruit of Rosa webbiana

The *n*-hexane, ethyl acetate and methanol extracts of fruit of *Rosa webbiana* were tested for toxicity in male Sprague-Dawley rats, but no death was reported in any case at a dose of > 5 g/kg (LD_50_ > 5.0 g/kg). Anti-epileptic effect of these extracts was investigated in the PTZ-induced chronic model of epilepsy. At the end of the 4-day protocol, the survival rate of animals in each group (n = 7) was recorded (Figure 1a). The maximum survival rate (80%) was observed in case of the *n*-hexane extract followed by the methanol extract (40%). No animal could survive in case of the PTZ and EtOAc extract. These *n*-hexane extracts were also evaluated for cytotoxicity in MCF-7 cells through an MTT reduction assay. As evident (Figure 1c), no significant decrease in cell survival fraction could be observed for the tested doses (0.5 mg/mL and 1 mg/mL), further supporting the safety of the *n*-hexane extract. Based on these observations, only the *n*-hexane extract was selected for investigating a dose-dependent effect (at 50, 150 and 300 mg/kg). The animals were divided into five different groups (n = 7), including PTZ, diazepam (positive control) and three doses of *n*-hexane extract. Almost 40 min after administration of the extract, a subconvulsive dose of PTZ (40 mg/kg) was administered to all the groups except control. The behavioral response was recorded as a seizure score. A dose-dependent effect was observed, 300 mg/kg being the most effective dose at *p* < 0.001 (Figure 1b). 

### 2.2. Anti-Oxidant Potential of the N-Hexane Extract of Fruit of Rosa webbiana

The anti-oxidant potential of *n*-hexane extract was also investigated in vitro through detecting the free radical (DPPH) scavenging potential. Ascorbic acid was used as a standard anti-oxidant agent. Free radical scavenging effect is evident in Figure 2a,b in terms of dose effectiveness and IC_50_ values. The *n*-hexane extract was found to possess anti-oxidant potential. 

### 2.3. Development of N-Hexane Extract into Chitosan Nanoformulation

The next step was the preparation of the nanoformulation of the of *n*-hexane extract using chitosan as carrier. For drug preparation, 30 mg/mL of the *n*-hexane extract was dissolved in sodium tripolyphosphate (TPP) solution and was then added to the chitosan solution with continuous stirring for 60 min. Chitosan-TPP NPs (cs-TPP NPs) and *n*-hexane chitosan nanoparticles (*n*-hexane cs-NPs) in liquid form are shown in Figure 3a,b, respectively. UV analysis of cs-TPP NP solution showed maximum absorbance at 210 nm range, while the *n*-hexane cs-NPs peak was at 210 nm and 212 nm, confirming the preparation of chitosan-TPP NPs. The *n*-hexane cs-NPs showed maximum absorbance at 274 nm, 282 nm and 285 nm (Appendix A). A total of 69% EE was calculated on 274 nm (Figure 4). Confirmation of encapsulation was done through FTIR spectroscopy. In FTIR spectra of chitosan and TPP, a broad peak at 3264 cm^−1^ was detected, indicating the presence of O-H groups (Figure 5a). Likewise, the presence of C-N and N-H bond stretches was also observed (2120 cm^−1^ and 1637 cm^−1^). In *n*-hexane cs-NPs, different peaks were observed in contrast to chitosan-TPP NP FTIR spectra in the regions of 3273, 2927, 2127, 1639, 1450, 1253 and 1084 cm^−1^, indicating the presence of different phyto-constituent in the extract. In the zeta analysis, the zeta potential remained in the range of 29–30 mv for all stocks, but size exceed 1:6 of TPP to chitosan ratio, while a size up to 408 nm was obtained when the CS to TPP ratio was kept at 3:1. It was witnessed that the TPP:cs ratio affected the size of nanoparticles. Polydispersity index (PDI) is another parameter which affects the stability and uptake of nanoparticles by the cell membrane. The range of PDI of all stocks was in the range of 0.06–0.6. Although the size enhanced after the encapsulation of the extract, zeta and PDI reduced to 4.62 and 0.069, respectively. Changes in the zeta potential after encapsulation of the plant extract are evident in Figure 6a,b.

### 2.4. Evaluation of Chitosan Nanoformulation for Anti-Convulsant Effects

Finally, the anticonvulsant potential of cs-NPs and *n*-hexane cs-NPs was determined using the same protocol as described above. Experimental groups included PTZ (40 mg/kg), Diazepam (4 mg/kg), cs-NPs control, *n*-hexane extract and *n*-hexane cs-NPs CS-P.E. NPs. Results in terms of survival rate and seizure score are given in Figure 7a,b. One-way ANOVA followed by Tukey’s multiple comparisons test indicated that PTZ significantly increased seizure score and mortality with recorded seizure score as 9.600 ± 0.1871 and 100% mortality (following seizures). However, continued administration of *n*-hexane extract and *n*-hexane cs-NPs resulted in a significant reduction in seizure score in the treated groups as follows: *n*-hexane (300 mg/kg) treatment group: 6.300 ± 0.6042 (** *p* < 0.01 relative to PTZ group), mortality, 40 ± 0 (** *p* < 0.01 relative to PTZ group) and *n*-hexane cs-NPs (300 mg/kg) treatment group: 5.300 ± 0.7000 (*** *p* < 0.001 relative to PTZ group), mortality, 20 ± 0 (*** *p* < 0.001 relative to PTZ group). These results demonstrated that both the *n*-hexane extract and *n*-hexane cs-NPs exhibited a statistically significant reduction in seizure score and mortality in both the treated groups as compared to the PTZ control group; however, results in terms of reduction in seizure score were more significant in case of encapsulation of the *n*-hexane extract than in the case of the *n*-hexane extract without encapsulation (Figure 7b). After 4 days, animals were sacrificed and their brains were preserved at −80 °C for further molecular analysis.

### 2.5. Effect of N-Hexane Extract and Its Nanoformulation on the Tnf-α and NF-KB Expression Detected Through Enzyme Linked Immunosorbent Assay

In order to examine the effects of *n*-hexane extract and *n*-hexane cs-NPs on PTZ induced neuro-inflammation and on the expression of NF-κB signaling pathway and a major cytokine TNF-α, cortex and hippocampus of rats were analyzed through ELISA. Two-way ANOVA followed by Tukey’s multiple comparisons test indicated (Figure 8a,b) a significantly upregulated NF-κB expression in the PTZ group ((cortex, 391.25 ± 12.67 (### *p* < 0.001 relative to saline group); (hippocampus, 329.75 ± 9.56 at *** *p* < 0.001 relative to saline group, respectively)). However, continued administration of *n*-hexane extract and *n*-hexane cs-NPs resulted in a significant downregulation in NF-κB expression in the treated groups, namely, n-hexane: cortex, 327.25 ± 6.556 at ** *p* < 0.01 relative to PTZ control group: hippocampus, 261.55 ± 5.334 at ** *p* < 0.01 relative to PTZ control group: *n*-hexane cs-NPs treatment group: cortex, 321.75 ± 8.789 at *** *p* < 0.001 relative to PTZ control group, hippocampus, 257.25 ± 12.576 at *** *p* < 0.001 relative to PTZ control group (Figure 8a). An important neuro-inflammatory cytokine downstream NF-κB is TNF-α. The expression of TNF-α was significantly upregulated in PTZ control group in both cortex and hippocampus: cortex, 3510 ± 80 (### *p* < 0.001 relative to the saline group); hippocampus, 3405 ± 90 (*** *p* < 0.001 relative to saline group), respectively. However, continued administration of *n*-hexane extract and *n*-hexane cs-NPs caused pronounced downregulation in TNF-alpha expression, namely, n-hexane: cortex, 2687 ± 70 at ** *p* < 0.01 relative to PTZ control group: hippocampus, 2720 ± 110 at ** *p* < 0.01 relative to PTZ control group and *n*-hexane cs-NPs: cortex, 1810 ± 80 at *** *p* < 0.001 relative to the PTZ control group, hippocampus: 1705 ± 90 at *** *p* < 0.001 relative to PTZ control group (Figure 8b). These results, therefore, indicate that *n*-hexane extract of fruit of *Rosa webbiana* reduced PTZ-induced neurodegeneration by decreasing NF-κB and downstream cytokine TNF-alpha expression in both the cortex and hippocampus of rats. Interestingly, *n*-hexane cs-NPs further reduced the expression of both molecular markers.

### 2.6. Cytoprotective Effect of N-Hexane Extract and Its Chitosan Nanoformulation on the Neurons in the Hippocampus and Cortex

Morphological differences in the cortex and hippocampus of the brain using hematoxylin and eosin (H&E) staining were also recorded in this study. A significant reduction in the number of surviving nuclei in PTZ group compared with that of saline group is evident in both cortex and hippocampus (Figure 9). However, *n*-hexane extract attenuated these damages, increased the number of surviving neurons, while *n*-hexane encapsulated cs-NPs further improved the number of surviving neuronal cells at *** *p* < 0.001 (Figure 9, graphs on the right). 

### 2.7. Detection of Constituents in N-Hexane Extracts of Rosa webbiana

The presence of multiple functional groups was identified through FTIR in n-extracts of fruit of *Rosa webbiana*. Stretching frequencies of C=O, C=C, C=N, S=O, O-H, N-H, C-H and C-O were observed at different frequencies as shown in Table 1. Apart from the presence of aldehyde, alkanes, amines and C-X functional groups, esters and acids were also detected through FTIR analysis (Figure 7b). To support further these findings, GC-MS analysis of *n*-hexane extract was also conducted which also detected the presence of long chain alkanes, (heptadecane, pentadecane, eicosane and heneicosane), heterocyclic amines (*N*-methylcaprolactam, 2, 4, bis (hydroxylamino) and 5-nitropyrimidine) and esters or acids (Table 2).

## 3. Discussion

The role of plants in protecting the brain cells in response to different kinds of insults inflicted to brain is highly interesting. Particularly, fruits are well reputed for their neuroprotective effects. For reference, literature can be cited where formulations with multiple plant components have shown to be therapeutically effective in CNS diseases [25]. *Rosa damascene* and *Rosa sinensis* have been reported for their anti-convulsant, hypnotic and neuroprotective effects [26,27]. *Rosa webbiana* is closely related to these species and grows in northern areas of Pakistan. Literature on its pharmacological and phytochemical investigations is very limited. During our survey, we found that the fruit is edible, safe and popular among the local children. In our study, on initial screening for anti-epileptic potential in PTZ induced model of epilepsy in rats, only *n*-hexane extract was selected for further investigation due to its safety profile (Figure 1). During a literature survey, we found that *n*-hexane extracts of multiple plants, constituting low to non-polar compounds, exert neuroprotective effects.

However, low bioavailability, rapid degradation, especially of flavonoids by enzymes or endogenous flora, and pH changes limit their bioavailability in vivo. The restrictions imposed by the BBB, a composition of endothelial cells connected to capillaries by tight junctions separating the main circulation from extracellular fluid of brain, limit the bioavailability of plant constituents [28]. There have been several recent reports demonstrating that nanocarriers in solid-lipid forms, polymeric encapsulations and liposomes have been successful in delivering the active fractions of plants through improved BBB permeation [29]. Chitosan (polymeric) encapsulation of *n*-hexane extract of fruit of *Rosa webbiana* reduced the seizure score, while survival rate was improved (Figure 8a,b). These results confirmed that nanocarriers of chitosan effectively delivered more drug to the brain crossing the BBB. Several important amino acids, including nitro-L-arginine and leucine, antidepressant compounds, nortriptyline and 2, 4-Bis (hydroxylamino)-5-nitropyrimidine were detected in this extract. However, the major peak belongs to an octyl ester (Table 1).

Two major factors involved in mediating neurological disorders and epilepsy are oxidative stress, damaged BBB and the resulting neuro-inflammation, as shown in Figure 10 [22]. Among the detected compounds in the fruit extract in our study, 1,2 benzenedicarboxylic diisooctyl ester, tetracosane, nitro-L-arginine and n-heneicosane are reported antioxidant agents from other members of family Rosaceae [22]. The observed antioxidant potential of *n*-hexane extract of *Rosa webbiana* fruit can be attributed to these constituents (Figure 2). As per the GCMS analysis, the major component was proposed to be dioctyl phthalate ester; however, this is just the proposed structure. The actual active ingredient can be some other molecule with the same molecular formula. Since *Rosa webbiana* fruit extract is rich in different phenolic acids, most probably, this peak may also correspond to some phenolic esters with the same molecular formula, and many pharmacological activities of this plant are due to the presence of these phenolic acids. Since *n*-hexane extract could not show toxicity in vivo or in vitro (Figure 1a,c), we suggest that this peak contributes to the reported pharmacological actions. 

An important factor involved in epileptogenesis is inflammation; strong evidence supports the notion that the two are interconnected [30,31,32]. NF-κB and TNF-α play a major role in the release of inflammatory cytokines and chemokines during epileptic seizures which aggravate the damage further [33]. The neuronal and other immune cells in the brain respond to exogenous and endogenous inflammatory inducers (through PAMPS and DAMP). On binding to specific receptors in cell membranes or in cytosol, these trigger cascade of events, stimulating NF-κB, MAPK, ERK, JNK and upregulating the expression of mediating chemokines and cytokines including IL-6, IFN, and TNF-α [34]. The reduced expression of NF-κB and TNF-α in PTZ induced animal model of epilepsy shows that the antiepileptic potential of fruit of *Rosa webbiana* may be the outcome of the attenuation of NF-κB and TNF-α (Figure 9a,b). Enhanced anticonvulsant activity and downregulation of NF-κB and TNF-α by chitosan-encapsulated *n*-hexane extract clearly shows that the lipophilic constituents present in this extract have been successfully delivered to the brain. The significant increase in the number of neurons in response to extract and further increase in case of nanoformulation, as shown in Figure 9, further confirms the cytoprotective effect of the fruit of *Rosa webbiana*. Other plants of this family have been studied for their neuroprotective potential, but this is the first report of the anticonvulsant effects of *Rosa webbiana*. GC-MS analysis has revealed esters, acids and amines as the main constituents, among others. This study warrants further investigation of this potential natural product on other neurodegenerative disorders, exploring further aspects through genomics and proteomics. 

## 4. Materials and Methods

### 4.1. Plant Material Collection and Extraction

Plant material consisting of fruity twigs, was collected from Shogran, on the Siri top, situated in the Himalayan range in Pakistan (Figure 11) in August and September of 2018. The plant along with its fruit was identified and preserved at the Herbarium of Pakistan, Quaid-e-Azam University, Islamabad, Pakistan (voucher specimen number 130285). Fruit was plucked, washed, dried, crushed into coarse powder (1 kg) and macerated using different solvent systems, including *n*-hexane, ethyl acetate (EtOAc) and methanol (MeOH), respectively, at room temperature for 14–15 days. The three extracts (*n*-hexane, EtOAc and MeOH) were then filtered and the solvent was removed using a rotary evaporator under reduced pressure. The yield obtained in n-hexane, EtOAc and MeOH was 40, 33 and 30 g respectively. All the extracts were stored at −4 °C until further use. 

### 4.2. Pharmacological Studies

#### 4.2.1. Toxicity Study

For initial screening for toxicity, n-hexane, EtOAc and MeOH extracts of fruit of *Rosa webbiana* were administered to the male Sprague-Dawly rats (250–300 g). The extracts (5 g/kg) were dissolved in normal saline: Tween 80 mixed with continuous stirring, until a homogenous mixture was obtained. The extracts were administered intraperitoneally. The observations were made after 72 h.

#### 4.2.2. Cytotoxicity 

The *n*-hexane extract of fruit of *Rosa webbiana* was subjected to the MTT reduction assay to investigate their growth inhibitory potential against breast cancer cells (MCF-7). The cell survival fraction values of *n*-hexane extract were calculated (n = 3), as shown in Figure 2c. The doses selected were 0 mg/mL, 0.5 mg/mL and 1 mg/mL. 

#### 4.2.3. Anticonvulsant Activity

All experimental procedures were set according to the protocols approved by Riphah Ethical Committee, Riphah Institute of Pharmaceutical Sciences (Ref. No. REC/RIPS/2019/29). Adult male Sprague–Dawley rats weighing 270–300 g were purchased from the National Institute of Health (NIH), Islamabad, Pakistan. The experimental animals were kept in an animal house at Riphah Institute of Pharmaceutical Sciences, under a 12 h dark/light cycle at 18–22 °C and had free access to water and food throughout the study. A pentylenetetrazole (PTZ)-induced model of epilepsy was followed in this study. The animals were divided into five groups (n = 7) as follows: normal saline; PTZ administered group, diazepam administered group; PTZ + extract (the *n*-hexane, EtOAc and MeOH extracts; 50 mg/kg); administered experimental rats. PTZ (40 mg/kg) was used as an inducing agent, while diazepam was used as standard drug (positive control). On the basis of survival rate, only *n*-hexane extract was selected for further study (Figure 2a) at three different doses, including 50 mg/kg, 150 mg/kg and 300 mg/kg (Figure 2b). Convulsive behavior was observed for 30 min after each injection of PTZ and resultant seizures were scored according to the following scale: score 0, no response; score 1, behavior arrest with trembling; score 2, motionless staring and sudden arrest in animal behavior; score 3, facial twitching; score 4, neck jerks; score 5, clonic seizure in sitting position; score 6, clonic seizure but animal do not lose its balance; score 7, tonic-clonic seizure; score 8, tonic-clonic with falling on one side; score 9, wild jumping; score 10, tonic extension leading to death [35]. After recording these observations, rats were sacrificed and their brains were preserved in formalin (10%) solution without formalin at −80 °C for further use.

#### 4.2.4. Detection of Expression of p-NF-kB and p-TNF-α through ELISA 

The changes in the expression of the levels of p-NF-κB and p-TNF-α were detected through ELISA in accordance with the manufacturer’s instructions (Shanghai Yuchun Biotechnology, Shanghai, China). The brain tissues stored at −80 °C were homogenized and the supernatant was collected on centrifugation (at 13,500× *g* for 1 h). The supernatant was used for quantifying the levels of p-NF-κB (Cat. No. SU-B28069, Shanghai Yuchun Biotechnology, China) and p-TNF-α (Cat.No. SU-B3098, Shanghai Yuchun Biotechnology, Shanghai, China). Briefly, the protein samples were reacted with respective antibodies provided in the kit using a 96-well plate and absorbance values were measured via microplate reader (BioTek ELx 808, BioTek Instruments, Inc. Winuschi, VT, USA). All experiments were performed in triplicate.

#### 4.2.5. Cytoprotective Effect Determined through Hematoxylin and Eosin (H&E) Staining

Brain tissues were fixed in formaldehyde 4% with PBS (0.1 M) and washed with water. Tissues were dehydrated by graded ethyl alcohol series from 70 to 100%, cleaned with xylene and fixed in paraffin using an embedding center (Leica, Westlar, Germany). Paraffin blocks were cut into 4 μm segments, deparaffinized with xylene and hydrated by graded ethyl alcohol series (from 100 to 70%). Segments were stained with Harris’ hematoxylin solution (Sigma-Aldrich, St. Louis, MO, USA) for 3 min and Eosin Y (Sigma-Aldrich) for 1 min. Segments were washed with water, dehydrated with graded ethyl alcohol series, mounted (Thermo Fisher Scientific, Waltham, MA, USA) and photographed using an Olympus microscope (Olympus, Tokyo, Japan).

### 4.3. Antioxidant Assay

An antioxidant assay of the *n*-hexane extract was performed using 1, 1-diphenyl-2-picrylhydrazyl (DPPH). Amount of discoloration of purple color showed the antioxidant activity of compound or extract [36]. The DPPH solution was prepared using methanol, while five different dilutions of extract using dimethyl-sulfoxide (DMSO) as solvent were prepared. Further dilutions were prepared using 100 µL from each previous dilution adding 500 µL of DPPH solution in each and incubated for 30 min in dark. After 30 min, the absorbance of samples was measured at a 517 nm wavelength using a Jasco-UV analyzer, while ascorbic acid was used as standard. The scavenging activity of the extract was determined using the following formula [37]:% Inhibition (DPPH) = Abs (control) − Abs (sample)/Abs (control) × 100(1)

### 4.4. GC-MS Analysis of Extract

GC-MS analysis of the *n*-hexane extract of *Rosa webbiana* was done using GC-MS QP 5050 A Gas Chromatograph Mass Spectrometer by SHIMADZU [37]. 

### 4.5. Synthesis of Chitosan Nanoparticles (cs-NPs)

The stock solution of chitosan (2 mg/mL) was prepared in 1% acetic acid solution, pH was maintained at 4.7–4.8. TPP stock solution (1 mg/mL) was also prepared using distilled water. Cs-TPP NPs were prepared using ionic gelation method [38]. To prepare *n*-hexane extract-NPs, the extract was dissolved in 3% DMSO and 2.5% Tween 80, which was added to the TPP solution and added dropwise to the chitosan solution After washing and centrifugation, nanoparticles were lyophilized and were stored at −20 °C for further studies.

#### Characterization of Nanoparticles

Nanoparticles were characterized to determine the amount of drug in particles, particle size, PDI and types of functional groups. Characterization was done using UV, FTIR and zeta analysis techniques. UV analysis of samples was done using (V-530 UV/VIS, JASCO, Int. Co. LTD) spectrophotometer at room temperature. Drug encapsulation efficiency was calculated using following formula:% Entrapment Efficiency = Actual drug content-free drug/Actual drug × 100 (2)

FTIR analysis of the *n*-hexane extract and its nanoparticles was done using the Alpha Bruker FTIR spectrophotometer (ATR eco ZnSe, vmax in cm^−1^). Liquid samples were mounted on the lens and the spectrum was generated by using OPUS software version 7. A Malvern zetasizer, which uses the Laser Doppler Electrophoresis Technique, was used to analyze the particles size, zeta potential and PDI of the nanoparticles in the samples.

### 4.6. Statistical Analysis

Data of ELISA were analyzed by a one-way analysis of variance followed by post-hoc Bonferroni multiple comparison tests (GraphPad Prism 6). Image J software was used to evaluate morphological data (H&E). Symbols # and * indicate *p* < 0.05, ## and ** indicate *p* < 0.01 and ### and *** indicate *p* < 0.001. Symbol # shows a significant difference compared to the saline group, while * shows a significant difference compared to the PTZ group.

## 5. Conclusions

In conclusion, we report here the neuroprotective effects of *n*-hexane extract of *Rosa webbiana*. This could be possible through the downregulation of epilepsy-induced raised levels of NF-κB and TNF-α. Overall, our results demonstrated that the effects of *Rosa webbiana* could be attributed to its enhanced anticonvulsant activity. The fruit of *Rosa webbiana* should be further investigated as a nutritional source and may lead to promising therapeutic agents in those patients who suffer from epilepsy mediated by ROS and inflammation. 

## Figures and Tables

**Figure 1 molecules-26-02347-f001:**
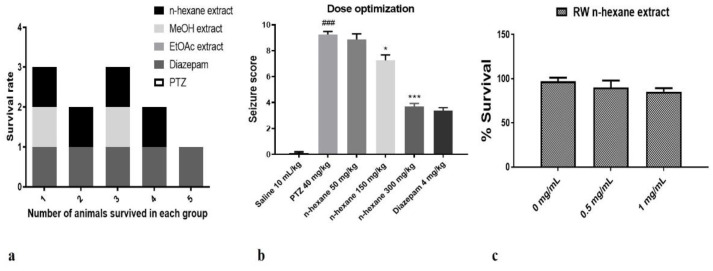
(**a**) Survival rate of animals (n = 5) in each group treated with PTZ, ethyl acetate, methanol and *n*-hexane extracts of fruit of *Rosa webbiana*. No animal could survive in case of ethyl acetate and PTZ, while two animals survived in methanol extract and four in *n*-hexane extract and diazepam. (**b**) The seizure score observed at 50 mg/kg, 150 mg/kg and 300 mg/kg of *n*-hexane extract. The data were expressed as mean ± SEM. Symbol * indicates a significant difference from the PTZ group at *p* < 0.05, while *** indicates significant difference at *p* < 0.001; symbol ^###^ shows a significant difference from saline at *p* < 0.001. (**c**) The cell survival fraction in MCF-7 cells (n = 3) in response to the *n*-hexane extract of fruit of *Rosa webbiana* (RW).

**Figure 2 molecules-26-02347-f002:**
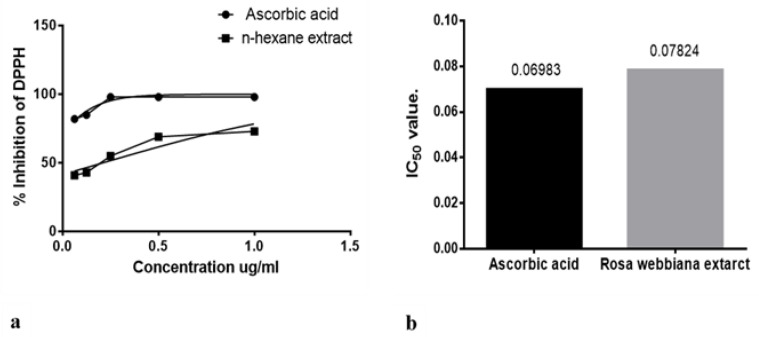
(**a**) DPPH scavenging by *n*-hexane extract of *Rosa webbiana* at different concentrations (**b**) IC_50_ value for ascorbic acid and *Rosa webbiana*. No significant difference from control (ascorbic acid) could be observed.

**Figure 3 molecules-26-02347-f003:**
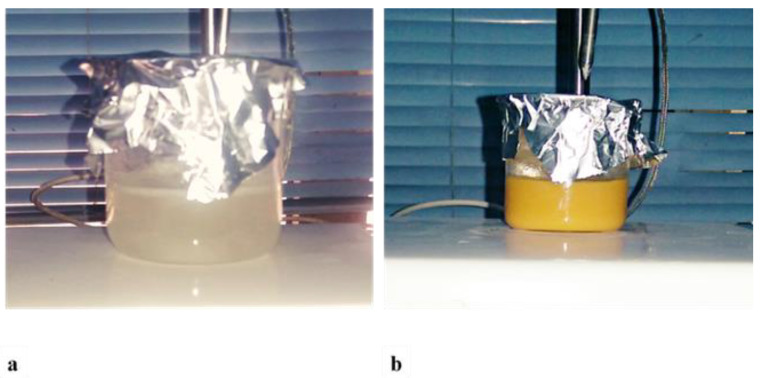
(**a**) Chitosan-TPP NPs. (**b**) Chitosan-P.E. NPs.

**Figure 4 molecules-26-02347-f004:**
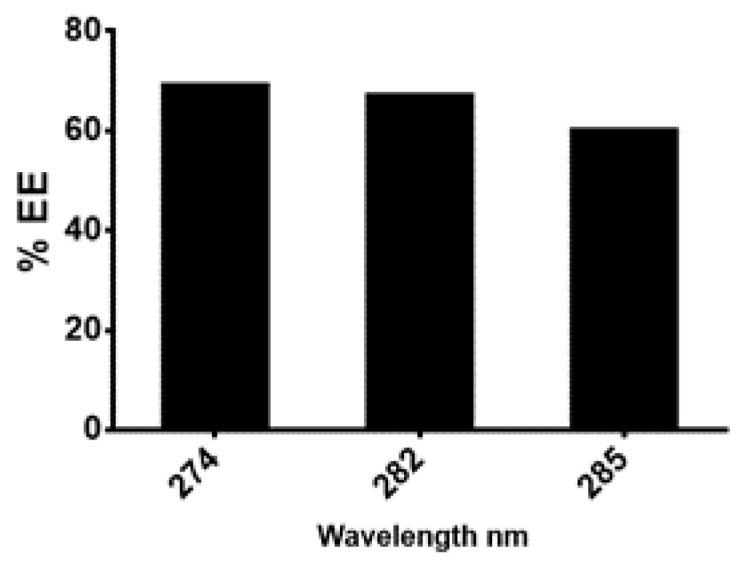
Percentage of the age encapsulation efficiencies calculated at different wavelength.

**Figure 5 molecules-26-02347-f005:**
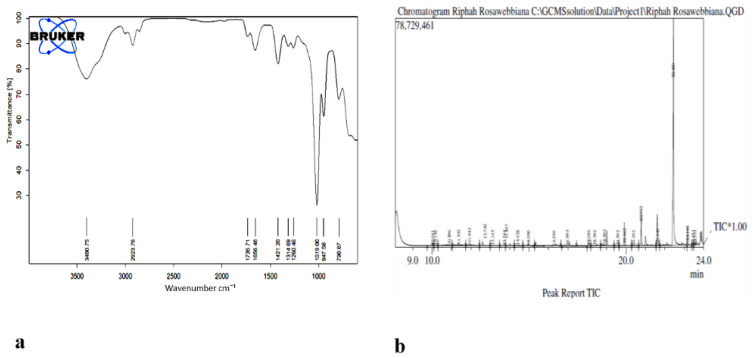
(**a**) FTIR spectra of *n*-hexane extract of *Rosa webbiana*. (**b**) GC-MS chromatogram of the *n*-hexane extract of fruit of *Rosa webbiana*.

**Figure 6 molecules-26-02347-f006:**
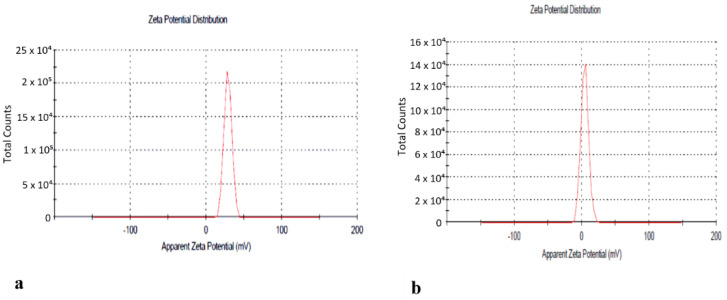
(**a**) Zeta potential of cs-TPP NPs. (**b**) Zeta potential of *n*-hexane cs-NPs.

**Figure 7 molecules-26-02347-f007:**
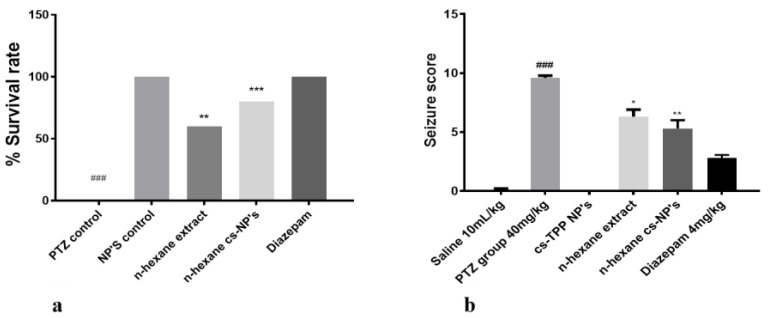
Illustrates (**a**) percent survival rate of all treated groups; symbols ** and *** show the significant difference from PTZ group at *p* < 0.01 and *p* < 0.001; symbol ^###^ shows significant difference from NPs control at *p* < 0.00. (**b**) Seizure score of all treated groups; symbols *, ** and ^###^ indicate significant differences at *p* < 0.05, *p* < 0.01 (from PTZ) and *p* < 0.001 (from normal saline), respectively.

**Figure 8 molecules-26-02347-f008:**
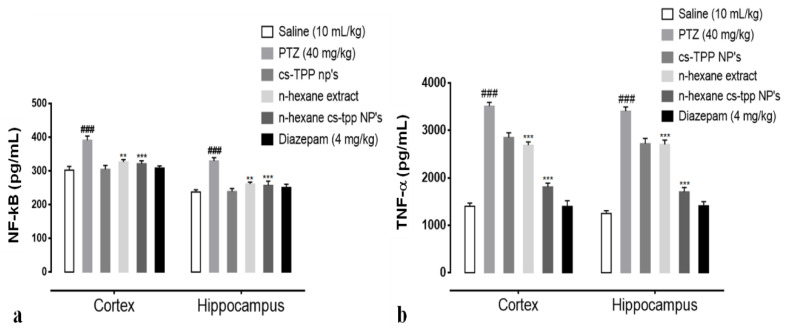
(**a**) Illustrates the effect of *Rosa webbiana* on the downregulated expression of p-NF-κB in the cortex and hippocampus of all treated rats; symbol ### symbolizes the significant difference from saline at p < 0.001, while symbols ** and *** indicate significant difference from PTZ at *p* < 0.01 and *p* < 0.001, respectively. (**b**) Effect of *Rosa webbiana* on TNF-α expression in the cortex and hippocampus of all treated rats; symbols. ^###^ symbolizes the significant difference from saline at *p* < 0.001, while symbols ** and *** indicate a significant difference from PTZ at *p* < 0.01 and *p* < 0.001.

**Figure 9 molecules-26-02347-f009:**
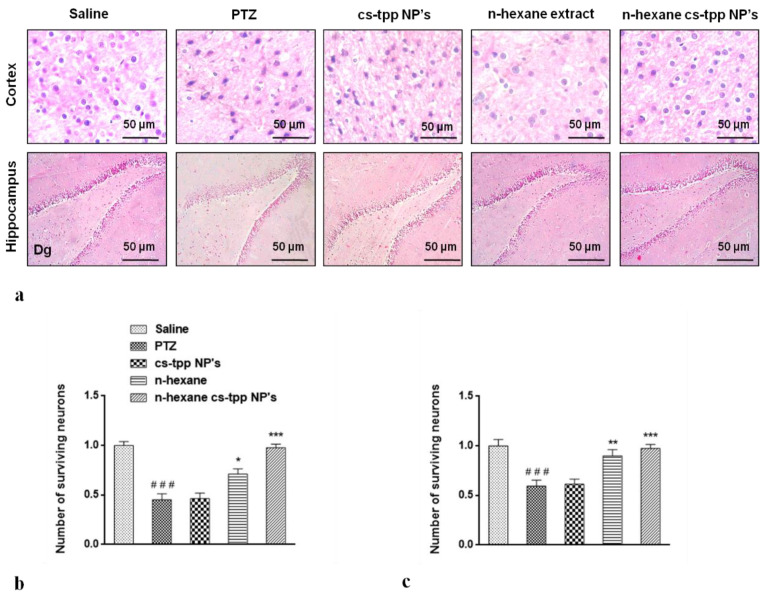
Slides illustrate morphological differences in the cortex and hippocampus of the brain of treated groups of animals using hematoxylin and eosin (H&E) staining (**a**). The effect on the number of neurons is evident in graphs; (**b**,**c**)Symbols ### indicate significant difference from saline at *p* < 0.001 while symbols *, ** and *** shows significant difference from PTZ (disease) group at *p* < 0.05, 0.01 and 0.001 respectively.

**Figure 10 molecules-26-02347-f010:**
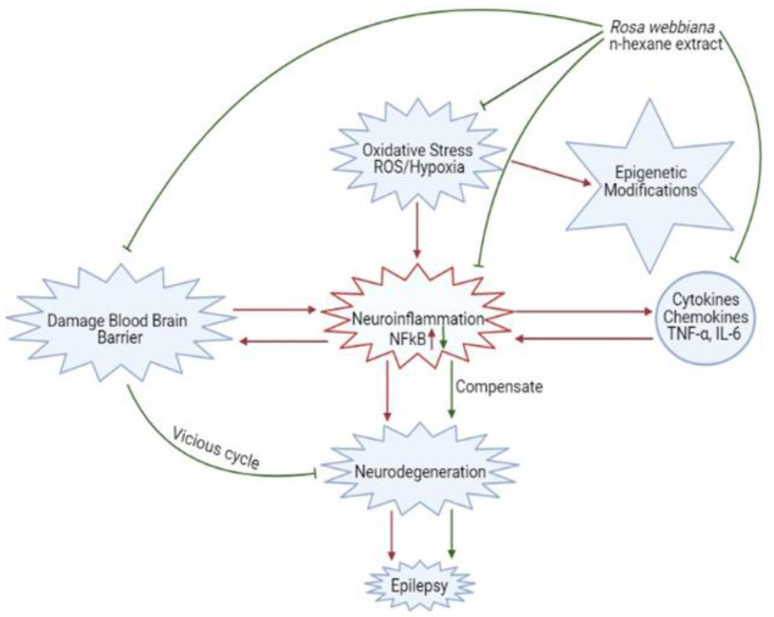
Schematic description of the contribution of oxidative stress, neuro-inflammation and blood–brain barrier destruction in epileptogenesis and the protective mechanisms of *Rosa webbiana*, including antioxidant mechanisms, neuroprotection through reduced cytokine release (TNF-α) and downstream NF-kB inhibition and improved drug delivery.

**Figure 11 molecules-26-02347-f011:**
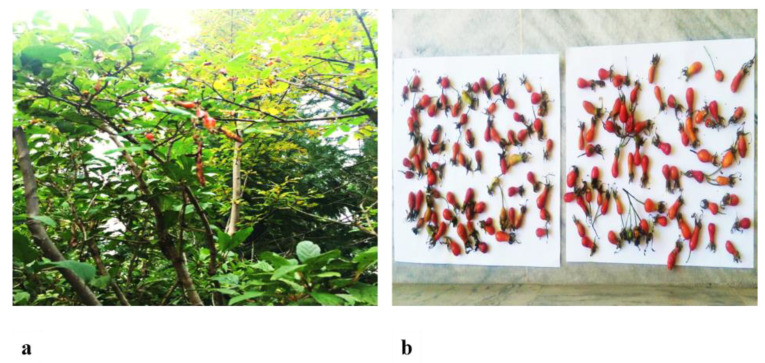
(**a**) *Rosa webbiana* plant. (**b**) Fruit.

**Table 1 molecules-26-02347-t001:** FTIR spectral data of *n*-hexane extract of fruit of *Rosa webbiana*.

Frequencies	Functional Groups
3400 cm^−1^	C=0, O-H, C-H and N-H of amines, alkanes and alcohols.
2923 cm^−1^	C-H, CH3 band of alkanes
1735 cm^−1^	C=0 of ester and acids
1656 cm^−1^	C=O, C=C of aryl ketones, arenes and alkenes.
1421 cm^−1^	CH2 group
1314 cm^−1^	C-N, C-O bonds
1260 cm^−1^	CH2 rocking and S=0 bond
790 cm^−1^	C-X bond

**Table 2 molecules-26-02347-t002:** Compounds detected through GC-MS analysis.

Peak #	R. Time	Constituents	Activity	Area%
1	11.392	Heptadecane, Pentadecane	-	2.28
2	11.942	Pentanoic acid or valeric acid	Neuroprotection, GABA-ergic effect	2
3	12.742	*N*-epsilon-methyl-*L*-lysine	Control transcription and translation and effect expression of proteins [20]	1.33
4	16.296	Nortriptyline	Neuroprotective, anti-inflammatory [21]	1.67
5	16.993	*N*-methyl epsilon-*C*-parolactam	-	1.34
6	18.905	Nitro-*L*-arginine	Amino acid, anti-oxidant and anti-inflammatory [22]	4.13
7	19.942	2,4-Bis (hydroxylamino)-5-nitropyrimidine	Anti-depressant, neuroprotective [22]	4.74
8	20.793	Tetracosane	-	9.24
9	22.450	1,2-Benzenedicarboxylic acid, di-iso octyl ester or some other ester	Cytotoxic, antioxidant and fungicidal effects [23,24]	55.66

## Data Availability

Not applicable.

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
