# Peer review of "Contribution of Attenuation of TNF-α and NF-κB in the Anti-Epileptic, Anti-Apoptotic and Neuroprotective Potential of Rosa webbiana Fruit and Its Chitosan Encapsulation"

_molecules, 2021, doi:10.3390/molecules26082347_

Round 1

Reviewer 1 Report

This paper deals with the anti-epileptic and neuroprotective potential of Rosa webbiana fruits and chitosan encapsulation. The author demonstrated the major chemical constituent, 1,2-benzenedicarboxylic acid di-iso-octyl esters up to 55.6%. Therefore, this chemical seems to be the major active principle for the biological assay through this manuscript. However, this chemical is known as a major plasticizer and often contaminates in the extract and fraction during the preparation process. This compound is not the natural compound of Rosa webbiana. Therefore, the author needs to evaluate those activities again with the extract without this plasticizer. This plasticizer is known as DOP, one of the major plasticizers and contaminants.

Reviewer 2 Report

Thank you for submitting the manuscript “Contribution of Attenuation of TNF-α and NF-κB in Anti-epileptic, Anti-apoptotic and Neuroprotective Potential of Rosa webbiana Fruit and its Chitosan Encapsulation” to Molecules.

1) Abstract: 2-benzenedicarboxylic acid di-iso-octyl ester is the compound responsible for anti-epileptic activity? I think this can be made clearer in the abstract

2) All text has grammatical, typing and punctuation errors. Please check all the text to improve the quality of the writing. Also check space between number and unit.

3) Introduction: What is blood purification? It is a disease?

4) Introduction: In general, the authors talk a lot about the use of plants with anti-epileptic function, but in the case of the species studied, only four sentences were added at the end of the introduction.

5) M&M: is only death indicative of toxicity? I believe that if the authors really wanted to prove that there was no toxicity, a human cell test would be necessary.

6) Results: “However effect was lower than ascorbic acid” I don't think this contributes to the work because it seems obvious that an isolated pattern will have a better effect than an extract.

7) Results: FTIR needs to be further discussed. I suggest including a table with the wavelengths and possible identifications.

8) In all figures, a description of the statistical analysis must be performed.

Round 2

Reviewer 1 Report

The author responded adequately to all of the requirements from the reviewers.

Reviewer 2 Report

Thank you for submitting the review to Molecules. The authors responded to all the suggestions in my review and the manuscript greatly improved its scientific quality. However, before final acceptance it is necessary to make a new revision as there are still formatting problems that cannot appear in the final publication, for example in Table 1 or on page 11, among others.